

# Tucuxi-BLAST: Enabling fast and accurate record linkage of large-scale health-related administrative databases through a DNA-encoded approach

José Deney Araujo[1], Juan Carlo Santos-e-Silva[1],
André Guilherme Costa-Martins[1,2], Vanderson Sampaio[3,4],
Daniel Barros de Castro[5], Robson F. de Souza[6], Jeevan Giddaluru[1],
Pablo Ivan P. Ramos[7], Robespierre Pita[7], Mauricio L. Barreto[7],
Manoel Barral-Netto[7] and Helder I. Nakaya[1,2,4,8]

[1] Department of Clinical and Toxicological Analyses, Universidade de São Paulo, São Paulo, SP, Brazil
[2] Scientific Platform Pasteur USP, São Paulo, SP, Brazil
[3] Fundação de Medicina Tropical Dr. Heitor Vieira Dourado, Manaus, Brazil
[4] Instituto Todos pela Saúde, São Paulo, SP, Brazil
[5] Fundação de Vigilância em Saúde do Amazonas, Manaus, Brazil
[6] Departamento de Microbiologia, Universidade de São Paulo, São Paulo, Brazil
[7] Oswaldo Cruz Foundation, Salvador, Brazil
[8] Hospital Israelita Albert Einstein, São Paulo, SP, Brazil

Corresponding author
Helder I. Nakaya, hnakaya@usp.br

## ABSTRACT

**Background:** Public health research frequently requires the integration of information from different data sources. However, errors in the records and the high computational costs involved make linking large administrative databases using record linkage (RL) methodologies a major challenge.

**Methods:** We present Tucuxi-BLAST, a versatile tool for probabilistic RL that utilizes a DNA-encoded approach to encrypt, analyze and link massive administrative databases. Tucuxi-BLAST encodes the identification records into DNA. BLASTn algorithm is then used to align the sequences between databases. We tested and benchmarked on a simulated database containing records for 300 million individuals and also on four large administrative databases containing real data on Brazilian patients.

**Results:** Our method was able to overcome misspellings and typographical errors in administrative databases. In processing the RL of the largest simulated dataset (200k records), the state-of-the-art method took 5 days and 7 h to perform the RL, while Tucuxi-BLAST only took 23 h. When compared with five existing RL tools applied to a gold-standard dataset from real health-related databases, Tucuxi-BLAST had the highest accuracy and speed. By repurposing genomic tools, Tucuxi-BLAST can improve data-driven medical research and provide a fast and accurate way to link individual information across several administrative databases.

## INTRODUCTION

The major goal of epidemiology is to identify the factors associated with health-related conditions and events in a given population. This goal can be achieved by integrating medical, sociodemographic, and health data, which are often stored in separate databases. Cross-referencing data records from different sources is advantageous to the study of co-infections (*Teixeira et al., 2019*), disease recurrences (*Aiona et al., 2015*; *Balieiro et al., 2021*), identification of transmission hotspots (*Workneh, Bjune & Yimer, 2016*), and household transmission of leprosy (*Teixeira et al., 2020*) and can assist in health-related decision making (*Ali et al., 2019*). Many countries such as England, Canada, Australia, New Zealand, and Wales have invested in successful initiatives, including building large centers for data integration and developing new strategies (*Trudeau, 2017*; *Elias, 2018*; *Eitelhuber et al., 2018*; *Chitty et al., 2020*). Similarly, Brazilian initiatives have been successfully applied to large government databases (*Pita et al., 2018*; *Barbosa et al., 2020*), such as the Center for Data and Knowledge Integration for Health (CIDACS). CIDACS currently has a database covering about 55% of the Brazilian population, with information on 114 million individuals obtained by integrating administrative and health data, while operating at an excellent level of security and privacy (*Barreto et al., 2019*). The integration of administrative databases is achieved by a method known as record linkage (RL). However, a major limitation to RL pertains to the high rates of typing errors, absence of information, and inconsistencies in identification data (*Sayers et al., 2016*).

RL methods are divided into two categories: deterministic and probabilistic. Deterministic techniques use a unique and individual key, such as a social security number or other government identification codes (*Barbosa et al., 2020*). Although they are easy to implement, cheap, and require low computational power, the absence of a common key prevents the use of the deterministic approach for linking a large number of databases (*Pita et al., 2018*; *Ali et al., 2019*). On the other hand, probabilistic approaches utilize the identification data in records, such as patient's full name, sex, race, place of birth, identity number, date of birth, *etc.*, to link data from the same individual across the database, and estimates the probability of records from different databases belonging to the same individual. The main advantage of probabilistic methods is the possibility of using incomplete and erroneous data to perform the RL. However, implementing such methods is often costly and requires trained professionals (*Sayers et al., 2016*).

Some recent attempts to perform RL between databases containing hundreds of thousands of records required great investment in computational infrastructure (*Harron et al., 2017a*). To decrease the number of comparisons and therefore the computational demand, some RL programs utilize a strategy known as blocking (*Enamorado, Fifield & Imai, 2019*). In this strategy, data are ordered and grouped using predefined blocks, such as the first name or date of birth. Then, the probabilistic RL is performed only among individuals belonging to the same group, significantly reducing the number of records compared. Although fast, the blocking strategy requires manual curation to define the appropriate identification keys/fields for blocking and often needs a phonetic algorithm
(*Enamorado, Fifield & Imai, 2019*; *Ali et al., 2019*). In addition, mistypes and inconsistencies in identification records limit the sensitivity of the blocking strategy.

Here, we present Tucuxi-BLAST (https://github.com/csbl-br/tucuxi_blast), a tool that performs RL between administrative databases in an efficient and fast manner while protecting the personal information of individuals. Our method is based on the basic local alignment search tool (BLAST) (*Altschul et al., 1990*), widely used in bioinformatics to compare large databases of biological sequences. Tucuxi-BLAST first translates the individual's data into DNA sequences using a codon wheel that dynamically changes over different runs without impairing the efficiency of the process. This encoding scheme enables on-the-fly data encryption, thus providing an additional layer of privacy during the linking process. The comparison of the DNA-encoded identification fields is then performed using BLAST, and machine learning algorithms automatically classify the final results. Similar to comparative genomics where genes from different genomes are compared to determine common and unique sequences (*Emms & Kelly, 2019*), Tucuxi-BLAST also allows the simultaneous integration of data from multiple administrative databases, without the need for complex data pre-processing. We tested Tucuxi-BLAST on large databases containing real cases of patients infected with different pathogens (tuberculosis–TB, meningitis–MEN, and HIV/AIDS–HIV) and one database comprising mortality data. Tucuxi-BLAST outperformed other open-source probabilistic RL tools frequently used in epidemiological studies in terms of accuracy and speed. Applying Tucuxi-BLAST to a simulated database containing 300 million records, we showed that its RAM consumption was 4 GB on average and the processing time compatible with other blocking and indexing methods. The repositioning of bioinformatics tools for RL holds a promising potential for epidemiological and medical studies.

## MATERIALS AND METHODS

### Implementation and code availability

Tucuxi-BLAST was developed in Python 3 programming language (v.3.7.3) and the RL between databases uses the BLAST algorithm (*Altschul et al., 1990*). The software was implemented on Linux but developed using a multiplatform language and libraries, so it can be easily adapted to run on any operating system such as Microsoft Windows and MacOS. Tucuxi-BLAST is available and regularly updated in the CSBL (Computational Systems Biology Laboratory, Los Angeles, CA, USA) repository on GitHub (https://github.com/csbl-br/tucuxi_blast).

### Data encoding to DNA sequences, preprocessing, and encryption

Four identification fields were used to perform RL in both the simulated and Brazilian administrative databases. Initially, Tucuxi-BLAST converts all characters from strings (*e.g.*, names and surnames) to upper-case and removes special characters and diacritics. In the main repository, we provide the Tucuxi-clean-data module, which performs the preprocessing separately from the main program. The identification fields were encoded into DNA sequences in the following order: (1) individual's name and surname; (2) date of

birth; (3) sex; and (4) mother's name and surname. Both query and reference (*i.e.*, subject) databases must be processed in the same manner.

To remove records that were duplicated in the databases, we developed Tucuxi-BW (available at https://github.com/csbl-br/tucuxi_blast) using Python and the cluster function of VSEARCH program (v.2.15.2) (*Rognes et al., 2016*). Additionally, to ensure data privacy, an encrypted key containing a random mixture of alphanumeric digits is generated on each Tucuxi-BLAST run. This key is used to encode the strings and numbers in the identification fields into DNA sequences.

## Sequence alignment using Tucuxi-BLAST

After encoding the identification records into DNA, we used the open source BLASTn algorithm (v.2.10.0) (*Altschul et al., 1990*) to align the sequences between databases. We used the following parameters in Tucuxi-BLAST: megablast and dc-megablast as a task; "-dust no"; "-max_hsps 1"; "-strand plus"; and "-evalue 1e−10". BLAST's default values were used for the remaining parameters of the search. The alignment of long sequences often generates score values (bit-score and E-value) greater than from the alignment of smaller sequences, even if the smaller sequences are identical to larger queries. Such a bias could penalize people with short names. To prevent such an effect caused by the size of the names in the analysis, the values of bit-score and E-value are represented as the percentage of the best possible hit for each sequence (*i.e.*, the sequence against itself). This normalization approach enables the comparison of sequences with different lengths.

## Simulated databases

To generate a simulated database containing records for 300 million Brazilians, we developed the Tucuxi-Curumim program (available at https://github.com/csbl-br/tucuxi_curumim). Using a list comprising the names and surnames of Brazilians obtained from the Brazilian Institute of Geography and Statistics (source: the 2010 official demographical census), this program generates records containing the following fields: names and surnames of the individual; names and surnames of the individual's mother; date of birth; and sex. To evaluate the processing capacity of Tucuxi-BLAST, we randomly selected 1, 10, 100, 1k, 10k, and 100k records from the 300M database (subject) and used them as queries. An in-house Python script was developed to introduce errors to 10% of the records in the simulated databases with more than 100 records. The introduced errors are often found in administrative databases and include the substitution and/or deletion of a single character in names/surnames or a digit in the date of birth, and missing surnames (*e.g.*, due to marriage). The simulated error rates were distributed as follows: 45% with random single-, double-, or triple-character substitution; 49% with middle-name deletion; 5% with one or more surname deletions; and 1% with completely blank field(s) (*e.g.*, the name of the mother). These error rates are similar to those found in the actual administrative epidemiological databases analyzed in this work. Finally, we increased the same number of records in sub-datasets (1, 10, 100, 1k, 10k, and 100k) that are not in the 300M database.

The inclusion was intended to insert noise from unmatched records in the subject database. The same datasets were used for the benchmark.

## Large administrative databases containing data on Brazilian patients

Access to the Information System for Notifiable Diseases (SINAN, *Sistema de Informações de Agravos de Notificação*) and the Mortality Information System (SIM, *Sistema de Informações sobre Mortalidade*) databases from the State of Amazonas, Brazil, were provided by Fundação de Vigilância em Saúde do Amazonas. Data usage for research purposes was approved by the Ethics Committee of the Fundação de Medicina Tropical Dr. Heitor Vieira Dourado, Amazonas, Brazil (Protocol no. 3.462.265). The SINAN databases (data from 2012 to 2017) pertaining to tuberculosis (TB), HIV/AIDS (HIV) and meningitis (MEN) include individual notifications accordingly. In the SINAN databases, deaths can be registered as caused by the disease (deaths reported as caused by TB, HIV, or MEN) or by other causes (such as motor vehicle traffic deaths, homicide, suicide, *etc.*). Because of the well-known underreporting of deaths in the SINAN databases, the SIM database (data from 2012 to 2018) that centralizes information on deaths was also included in the analysis.

Using the Shiny Framework (v. 1.7.1), we also developed the Tucuxi-Tail platform (https://tucuxi-tail.csbiology.org/), which facilitates a visual inspection of the record linkage results obtained. This platform first displays identification information from two given records that are potentially the same (herein defined by running Tucuxi-BLAST without E-value cutoff and retrieving all hits). Then, users can decide if these two records are the same entity in both databases (*i.e.*, whether they are correctly linked) or not. The Tucuxi-Tail platform was utilized to quickly curate 3,000+ pairs of records, thus generating the gold-standard dataset used in our benchmark.

## Classification models in machine learning

We applied machine learning algorithms for building classification models to optimize the RL and reduce the false-positive rate of BLAST alignment results. Using Tucuxi-Curumim, we created 75,000 simulated records, which were then distributed into four datasets: training_query, training_subject, testing_query, and testing_subject. Matched records (class 1) were generated by duplicating the records in both the query and subject datasets. Unmatched records (class 0) were records that were uniquely found in a given dataset. Each dataset contained 25,000 records, of which 12,500 were class 0 and 12,500 were class 1. To the class 1 records, we attributed an error rate of 15% (see the "Simulated databases" section). The classification models were based on Random Forest (RF) and logistic regression (LR) using the training sets (namely the training_query and training_subject datasets). The models were then tested based on their linkage of the testing_query and testing_subject datasets. We used the RF (n_estimators = 75; criterion = entropy) and LR (default parameters) algorithms from the scikit-learn package (*Pedregosa et al., 2012*).

The classification model was carried out using the BLAST metrics drawn from the comparison between the query sequence (dataset A) and the subject sequence (dataset B). The metrics utilized in the model were (1) normalized bit-score, (2) mismatch (number of

mismatches), (3) gapopen (number of gap openings), (4) qcovhsp (query coverage per high-scoring pairs), (5) sstart (start of alignment in subject), and (6) qstart (start of alignment in query). The normalized bit-score was calculated by dividing the bit-score value between the query and the subject by the best possible bit score value for that same query (*i.e.*, the bit score value between the query and itself).

## Computational structure and RL benchmarks

All analyses were performed on a Linux workstation, Intel Core i7-8700, with 32 GB. The benchmark was implemented with five other RL tools using the default parameters: Python Record Linkage Toolkit (*De Bruin, 2019*), RecordLinkage (*Sariyar & Borg, 2010*), fastLink (*Enamorado, Fifield & Imai, 2019*), free-version of Dedupe (*Gregg & Eder, 2019*), and CIDACS-RL (*Barbosa et al., 2020*). When blocking method was used, the individual's name and surname were employed for the blocking.

# RESULTS AND DISCUSSION

## The Tucuxi-BLAST approach

We repurposed the BLAST algorithm to efficiently handle the RL between large administrative databases. An overview of the Tucuxi-BLAST workflow is summarized in Fig. 1A. The first step initiated by Tucuxi-BLAST is to convert the identification records of all patients into DNA sequences. This is achieved using a codon wheel that converts each letter and number from the identification fields in a record into a codon (Fig. 1B). The codon wheel was designed to be dynamically controlled by a key, which can be any string of letters or numbers. The encrypted key spins the codon wheel without altering the position of the characters and numbers to be encoded. Since keys are randomly generated on each Tucuxi-BLAST run, the RL from one run cannot be directly used to decode further runs. Results can be reproduced if the same key is manually set during the RL process. RL software is usually not designed for encrypting sensitive personal information. Although the encryption provided by our method was not designed to be completely secure, Tucuxi-BLAST offers an extra layer of data protection making sensitive personal less "readable" (Fig. 1B).

Once the identification records are converted into DNA sequences, the BLASTn algorithm is used to establish the alignment between a query dataset and a subject dataset. To classify BLAST alignments (Fig. 1C) as matched (class 1) or unmatched (class 0), we used classification models built with machine learning algorithms (see Methods). We also developed a module, named Tucuxi-BW, that can be applied to a single dataset (Fig. 1D). The goal of Tucuxi-BW is to detect and remove duplicated records from a database.

## Tucuxi-BLAST's performance on a simulated database with 300M records

To demonstrate the robustness, speed, and accuracy of Tucuxi-BLAST, we created over 300,000,000 simulated records (subject database) containing the same combinations of names, surnames, dates of birth, and sex information commonly found in real databases (see Methods and Fig. 1A). From the subject database, we created six query datasets with

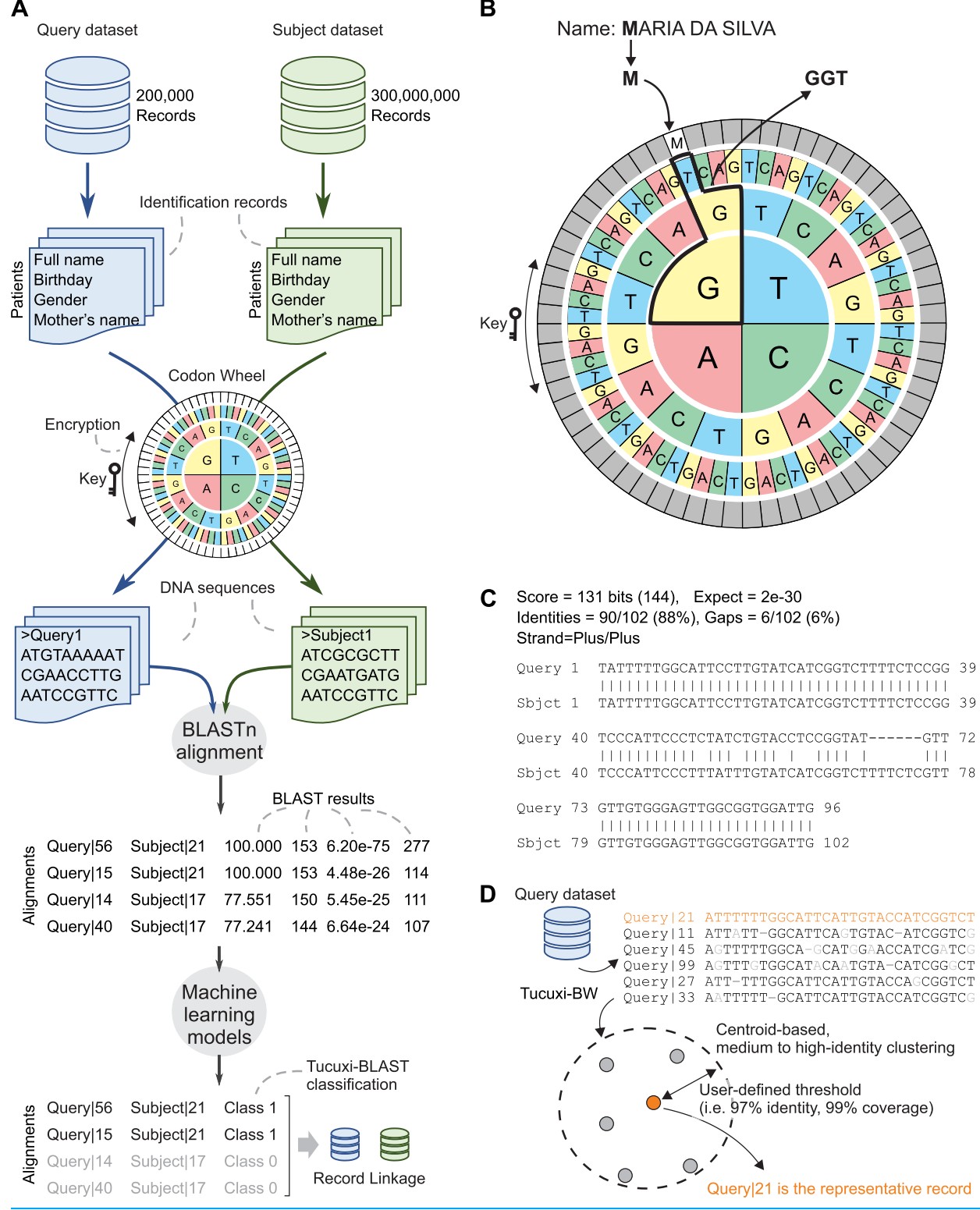

**Figure 1** **Tucuxi-BLAST workflow and data organization scheme.** Four variables are selected in common between two datasets, then DNA coding is performed. The coding result is submitted to the BLAST algorithm and, finally, ML is applied to classify the RL (A). Codon wheel used in DNA coding (B), results of BLAST for RL (C), and Tucuxi-BW module for data deduplication (D).

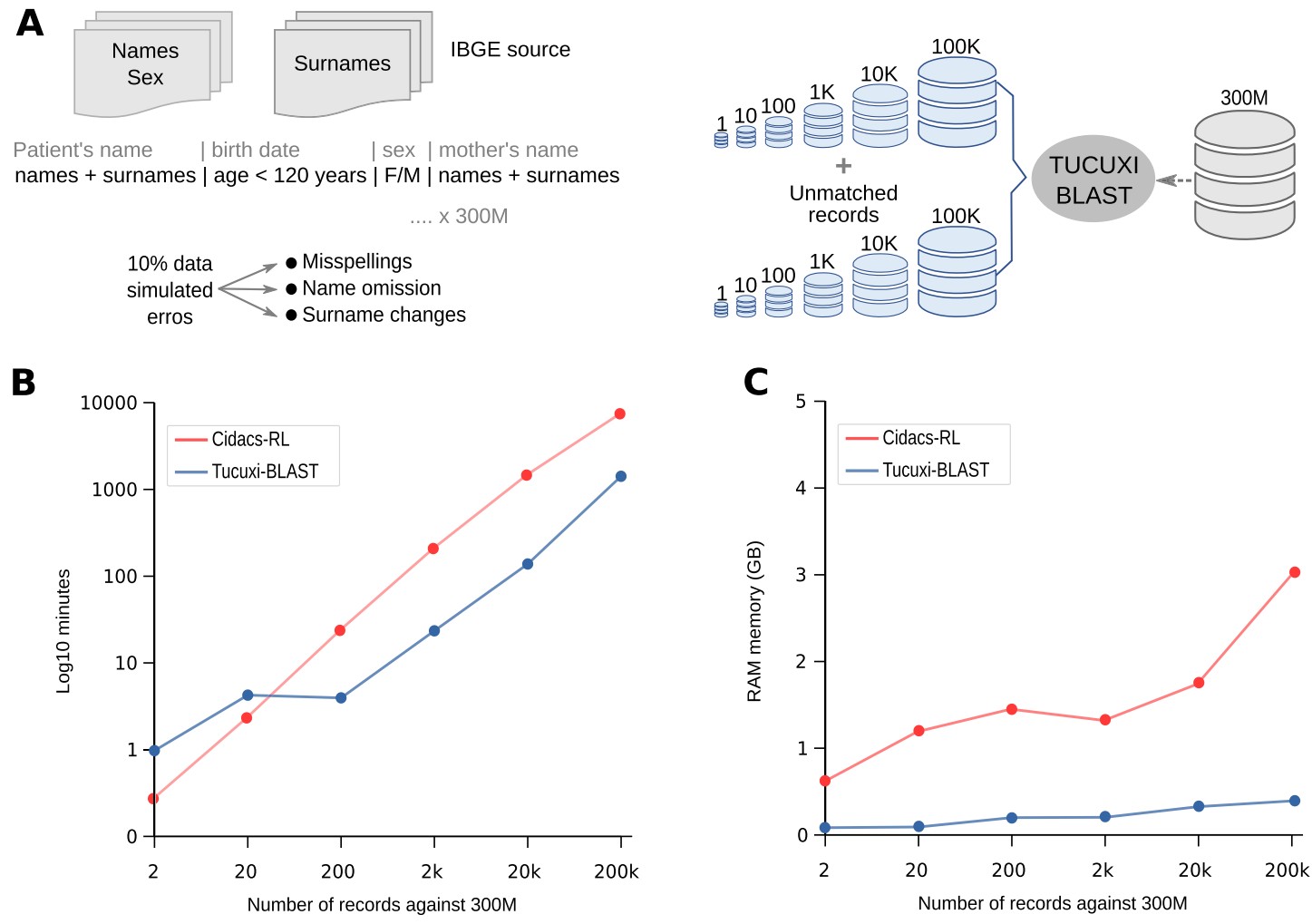

**Figure 2 Competence in handling big data.** Tucuxi-Curumim was used to generate all simulated data with the data obtained from IBGE (Instituto Brasileiro de Geografia e Estatística) (A). The execution time and use of RAM memory for each RL simulation were evaluated (B and C, respectively). All simulations were performed on a 32 GB Intel Core i7-8700 Linux workstation.                              

sampled records of different sizes (1, 10, 100, 1k, 10k, and 100k). For each query-simulated dataset, we randomly introduced misspellings, name omissions, and surname changes in 10% of the records. To evaluate the processing speed and accuracy of our method in dealing with true negative records, we also added in the six query datasets the same number of records that had no correspondence to the subject database (Fig. 2A). It has been shown that the presence of true negative records significantly increases the processing time for RL algorithms (*Harron et al., 2017b*).

Using the six query datasets containing 2–200k records against the reference subject database comprising 300M records (Fig. 2A), we performed a benchmark with Tucuxi-BLAST and five other RL methods (see Methods). Using the same PC with 32 GB of RAM, we could only run CIDACS-RL and Tucuxi-BLAST due to the limited RAM available. Before the linkage, both methods first created an index of the subject database. Tucuxi-BLAST took ~55 min and used 3.8 GB of memory to convert the entire dataset into

*in silico* DNA sequences to create the indexes. CIDACS-RL took ~80 min and used 1.7 GB to compile the indexes. Performance was then assessed by monitoring the processing time and RAM usage. Both methods' processing times were similar when query datasets included 20 records or less (Fig. 2B). In processing the RL of the largest query dataset (200k records), Tucuxi-BLAST was 5.69 times faster than CIDACS-RL. For this dataset, while CIDACS-RL took 5 days and 7 h to perform the RL, Tucuxi-BLAST only took 23 h (Fig. 2B). In terms of RAM usage, the maximum consumption of Tucuxi-BLAST and CIDACS-RL were 0.4 GB and 3 GB, respectively (Fig. 2C). These results are impressive and demonstrate the speed and accuracy of the platform. Currently, discussions on applying RL to big data have been raised although limitations on processing time are a concern. Tucuxi-BLAST is clearly aligned with these approaches and allows researchers with limited access to powerful servers or cloud services to perform RL in huge datasets in a timely manner.

## Tucuxi-BLAST applied to real administrative databases

We also performed RL on real administrative health-related databases from the Amazon State. Four databases were assessed: three query databases related/pertaining to tuberculosis (TB), HIV, or meningitis (MEN), and one subject database (SIM) dealing death registers. The goal was to try to identify patients in the query databases who were reported dead from a disease in the subject database.

For the benchmark, we used the Tucuxi-Tail platform to manually create a gold-standard dataset that contains deaths registered in both the query and subject databases. We found 2,382 individuals who were registered as dead in the query databases (MEN–203; HIV–936; and TB–1,243). Of those, 2,183 were also registered in the SIM database (91.6%). The remaining records (199) did not present a corresponding entry in the SIM database. Thus, for evaluating the RL, our gold-standard dataset utilized 2,183 true positive and 199 true negative cases.

We then used the true positive cases to investigate the errors between the query and subject databases. For all query databases, most errors were found in the mother's name (Fig. 3A). Besides typographical errors, the middle and last names of the patient's mother were often omitted or substituted by the husband's family name. Our results corroborate a previous report that showed that, due to marriage, the field corresponding to the mother's name has more mistakes between two linked records (*Dusetzina et al., 2014*). If all identification records were taken into account, a great proportion of patients (1,040 of linked records out of 2,183) had records with at least one mismatch/indel in the BLAST results (Fig. 3B). The MEN database showed the highest rate of records with at least one error compared with that of HIV and TB (Fig. 3B). The higher error rate in the MEN database may be due to several patients being newborns with meningitis. The names of these newborns usually change from the moment they are registered in the MEN database (which can be before the official birth registration) till they die (which is later registered in the SIM database). We also calculated the most frequent errors in the digits found in dates of birth. The most frequent errors were related to number-switches between the numbers "1" and "0" (0.68%), "5" and "6" (0.60%), and "6" and "7" (0.53%) (Fig. 3C). Similarly, we
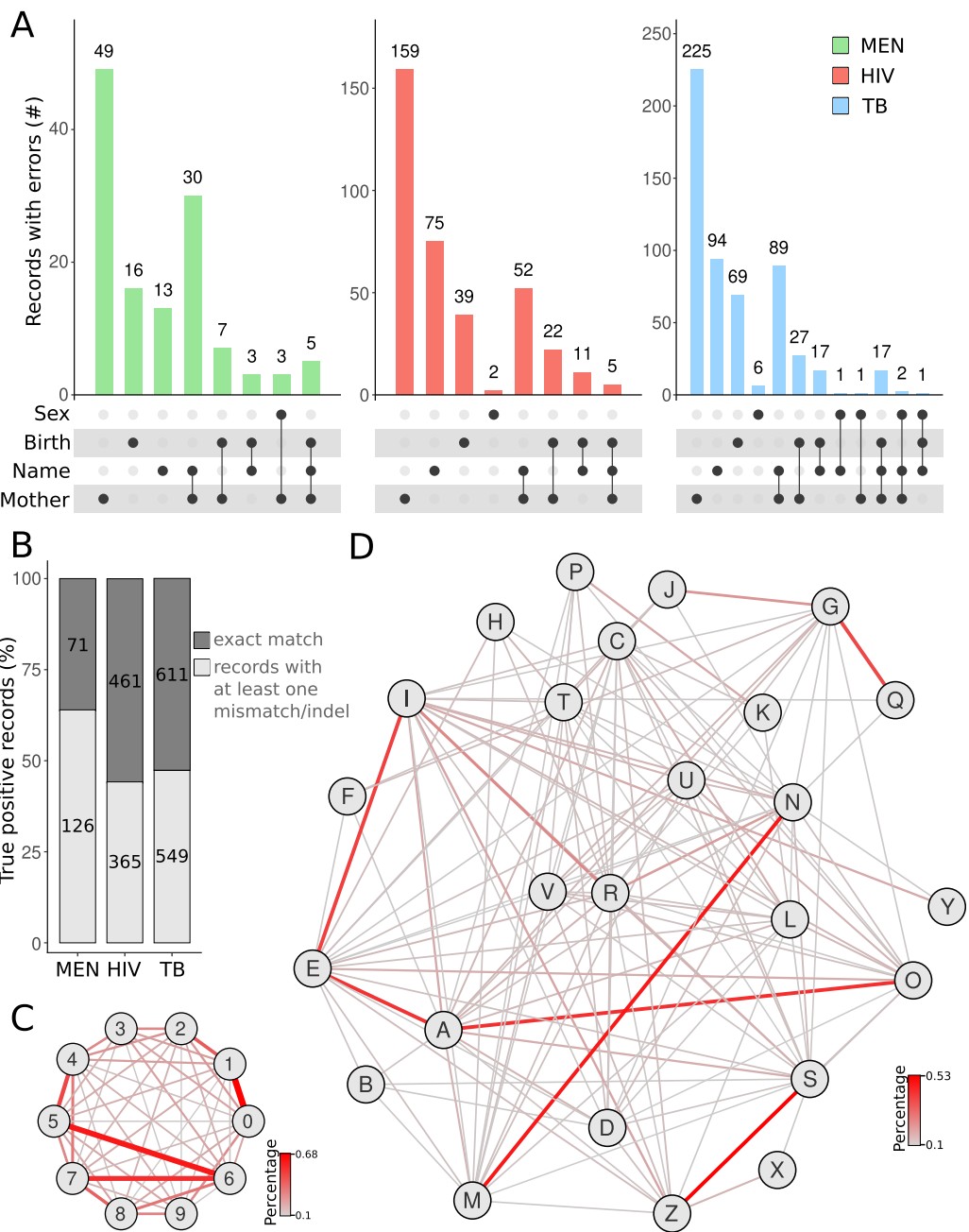

**Figure 3 Exploration of databases.** Counting the number of records with errors and in which variables the errors occur (A). Total error rates in true positive linked records of SINAN databases against SIM mortality databases identifying any type of error (B). The networks demonstrate the substitution rate between numbers (C) and letters (D). The substitution rate between alphanumeric characters was calculated using records showing only mismatches in the BLAST results, *i.e.* fields from both records having the same length. The networks display the characters (nodes) and the frequency of substitutions between them (edges).

identified that the letters that were mostly switched in the names of true positive records were between "N" and "M", "S" and "Z", "G" and "Q", and "I" and "E" (Fig. 3D). In some cases (*e.g.*, the switch between "N" and "M"), the switching is a typo caused by the

proximity of the aforementioned letter keys in a keyboard. For other cases, this is due to the use of letters that share similar phonetics in Brazilian Portuguese language. The codon wheel can be easily adapted to the English language (or any other language). For this, the corresponding positions for each character in the codon wheel must take into account the phonetics and typos frequently found for the target language.

The analysis of the most frequent errors between the linked records was then performed to improve the codon wheel that converts record information to DNA sequences (Fig. 1B). We leveraged the degeneracy of the genetic code (*Crick et al., 1961*) so that the letters that are commonly mistaken differ in only one nucleotide in the codon. For example, whereas the difference between "V" (codon GAG) and "Z" (codon ATC) differs in all three nucleotides, the difference between "S" (codon ATA) and "Z" (codon ATC) differs in only one nucleotide. Thus, names commonly written as "Luiz" and "Luis" (pronounced the same way in Portuguese) will have a more similar DNA sequence than "Luiz" and "Luiv". "Luiv" is not a common name in Brazil. Therefore, the codon wheel improved the scores of the BLAST algorithm in the cases of substitution involving phonetically similar letters, which are very common in administrative datasets. This approach adapted for Portuguese, requires simpler and faster preprocessing steps compared with the phonetic SOUNDEX method, originally developed for the English language (*Jordão & Rosa, 2012*; *Marcelino, 2015*). In fact, some RL tools rely on the preprocessing steps for applying phonetic-based algorithms (*Camargo & Coeli, 2015*; *Enamorado, Fifield & Imai, 2019*). Errors related to keyboard typos, such as switching the letter "M" with the letter "N" (which are close to each other on the keyboard) are also frequent. Our method treats such typos the same way it does for phonetically similar letters. It uses codons with only one nucleotide difference to each letter.

The BLAST algorithm (*Altschul et al., 1990*) is optimized to handle sequences that do not align perfectly with each other because most biological sequences carry genomic variations (*e.g.*, indels and SNPs) between individuals. Such differences are even bigger when sequences from two different species are compared. Also, the algorithm that compares nucleotide sequences (*i.e.*, BLASTn) is very efficient in analyzing billions of sequences in terms of precision and speed (*Perkel, 2021*). By leveraging the BLASTn algorithm, Tucuxi-BLAST can quickly compare millions of records that are not identical and achieve great accuracy and precision.

## Benchmarking Tucuxi-BLAST against other RL tools

Since only our method and CIDACS-RL were able to run on the large simulated databases (Fig. 2), we compared the performance of Tucuxi-BLAST and other RL tools using the manually curated gold-standard actual administrative databases. While the biggest simulated database had 200k records linked to a database with 300M records, the real administrative databases contained only 2,382 records in total to be linked to a mortality database with 106,613 records. All RL tools employed were open-source and required neither high-performance computing infrastructure nor extensive preprocessing steps. All RL algorithms (with the exception of Tucuxi-BLAST and RL Dedupe) were derived

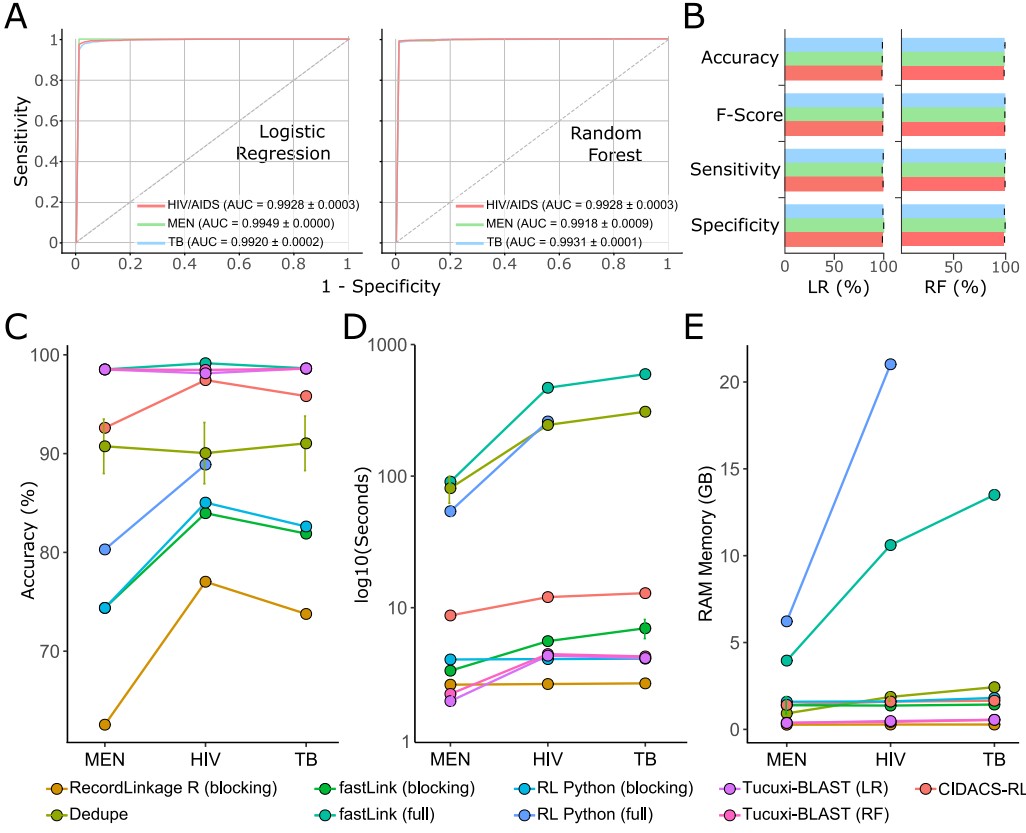

**Figure 4 Benchmark for the main record linkage tools.** ROC curves for the linkage runs of real data from disease databases of meningitis (MEN), HIV and tuberculosis (TB) using LR = Logistic Regression and RF = Random Forest (A). Performance metrics for the ML approach for each database (B). Accuracy percentage for each disease against death database using the different methods for the benchmark (C). Execution time spent (in log10 s) (D). RAM memory consumption in GB (E). Runs for the RecordLinkage R package applying non-blocking methods were not possible for the TB database using the workstation mentioned in the methods.

from the Jaro-Winkler method (*Winkler, 1990*) and/or the Damerau-Levenshtein method (*Levenshtein, 1966*). We used the same computer to run all the algorithms.

For the benchmark, we used the receiver operating characteristic (ROC) curve and the F-score to evaluate the performance of all RL methods. The RL was performed according to the documentation of each method, and the default parameters were used. Tucuxi-BLAST achieved an F-score and AUC above 98% for all three databases (Figs. 4A and 4B). Random Forest and logistic regression classifiers had a mean accuracy of 98.68% and 97.76% respectively (Table S1). Tucuxi-BLAST exhibited processing performance similar to the methods that use blocking; however, the accuracy was higher (Figs. 4C and 4D). The fastLink program, together with Tucuxi-BLAST, obtained the best performance (Fig. 4C). However, when considering the processing time, Tucuxi-BLAST is 100 times faster and consumes less memory than fastLink (Figs. 4D and 4E). In general, the worst performance was observed for programs that use the blocking approach. To speed up processing, blocking methods must reduce the number of comparisons between databases, which compromise the RL results (*Sayers et al., 2016*). Approaches combining blocking

and full-linkage methods were recently implemented. This approach has been shown to be efficient in many studies due to good performance metrics and low computational consumption (*Pita et al., 2018*; *Barbosa et al., 2020*). Although CIDACS-RL uses such a mixed approach incorporating both blocking and full-linkage methods, Tucuxi-BLAST was not only faster than CIDACS-RL but also demonstrated increased performance (Fig. 4).

The main limitation to using Tucuxi-BLAST for databases containing more than 300M records is the temporary storage on hard disk. DNA encoding and the BLAST software processing require temporary storage of at least twice the sum of the query and subject databases. In this case, if entries in both databases add up to a total of 5 GB, Tucuxi-BLAST will require at least 10 GB of hard disk space to proceed with RL. Despite the limitation concerning temporary storage, Tucuxi-BLAST scales well in terms of the usage of RAM memory (Fig. 2C). As the cost per gigabyte for hard disks continues to drop and the availability of large disks (>1 Tb) for general users is widespread, we believe that solving the storage allocation problem is much easier than dealing with the high RAM demand. Such low RAM demand makes Tucuxi-BLAST an effective long-term solution for the record-linkage problem, which is expected to keep increasing as new databases are continuously introduced.

## CONCLUSIONS

Our findings showed a high-resolution performance in record linkage by using an *in silico* DNA encoding system and the BLAST algorithm. The developed program was able to overcome misspellings and typographical errors in administrative databases. Tucuxi-BLAST does not require the installation of any dependencies, thus dispensing any prior knowledge of software management on Linux systems. Moreover, our approach, when compared with existing solutions, has clear advantages in terms of time and accuracy, even when using only four identification fields. In addition, the DNA encoding system introduces a layer of protection for securing personal information, helping to ensure confidentiality throughout the RL process.

## ACKNOWLEDGEMENTS

We are thankful to the members of Computational Systems Biology Laboratory (CSBL) for their help in revising the manuscript.

### Funding

This study was supported by grants from the Brazilian agency Fundação de Amparo à Pesquisa do Estado de São Paulo-FAPESP (grant No. 2018/14933-2) to Helder I. Nakaya. André Guilherme da Costa-Martins was the beneficiary of postdoctoral fellowships from CAPES (PNPD). Juan Carlo Santos e Silva received a doctoral fellowship from FAPESP (grant No. 2019/27139-5). The funders had no role in study design, data collection and analysis, decision to publish, or preparation of the manuscript.

## Grant Disclosures

The following grant information was disclosed by the authors:

Brazilian agency Fundação de Amparo à Pesquisa do Estado de São Paulo-FAPESP: 2018/14933-2.

CAPES (PNPD).

FAPESP: 2019/27139-5.

## Competing Interests

Helder I. Nakaya and Robson Souza are Academic Editors for PeerJ.

## Author Contributions

- José Deney Araujo conceived and designed the experiments, analyzed the data, prepared figures and/or tables, authored or reviewed drafts of the article, and approved the final draft.
- Juan Carlo Santos-e-Silva conceived and designed the experiments, analyzed the data, prepared figures and/or tables, authored or reviewed drafts of the article, and approved the final draft.
- André Guilherme Costa-Martins conceived and designed the experiments, analyzed the data, prepared figures and/or tables, authored or reviewed drafts of the article, and approved the final draft.
- Vanderson Sampaio performed the experiments, authored or reviewed drafts of the article, and approved the final draft.
- Daniel Barros de Castro performed the experiments, authored or reviewed drafts of the article, and approved the final draft.
- Robson F. de Souza conceived and designed the experiments, authored or reviewed drafts of the article, and approved the final draft.
- Jeevan Giddaluru analyzed the data, prepared figures and/or tables, authored or reviewed drafts of the article, and approved the final draft.
- Pablo Ivan P. Ramos conceived and designed the experiments, authored or reviewed drafts of the article, and approved the final draft.
- Robespierre Pita conceived and designed the experiments, authored or reviewed drafts of the article, and approved the final draft.
- Mauricio L. Barreto conceived and designed the experiments, authored or reviewed drafts of the article, and approved the final draft.
- Manoel Barral-Netto conceived and designed the experiments, authored or reviewed drafts of the article, and approved the final draft.
- Helder I. Nakaya conceived and designed the experiments, analyzed the data, prepared figures and/or tables, authored or reviewed drafts of the article, and approved the final draft.

## Human Ethics

The following information was supplied relating to ethical approvals (*i.e.*, approving body and any reference numbers):

Ethics Committee of the Fundação de Medicina Tropical Dr. Heitor Vieira Dourado, Amazonas, Brazil (Protocol no. 3.462.265).

## Data Availability

Tucuxi-BLAST and the simulated datasets containing 300M (subject) and 200M (query) are publicly available and regularly updated in the CSBL (Computational Systems Biology Laboratory) repository at GitHub: https://github.com/csbl-br/tucuxi_blast.

## Supplemental Information

Supplemental information for this article can be found online at http://dx.doi.org/10.7717/peerj.13507#supplemental-information.

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

# PeerJ

**De Bruin J. 2019.** Python record linkage toolkit: a toolkit for record linkage and duplicate detection in python. Zenodo DOI 10.5281/zenodo.3559043.

**Dusetzina SB, Tyree S, Meyer AM, Meyer A, Green L, Carpenter WR. 2014.** *Linking data for health services research: a framework and instructional guide.* Report No.: 14-EHC033-EF. Rockville: Agency for Healthcare Research and Quality. *Available at* https://www.ncbi.nlm.nih.gov/books/NBK253313/.

**Eitelhuber TW, Thackray J, Hodges S, Alan J. 2018.** Fit for purpose-developing a software platform to support the modern challenges of data linkage in Western Australia. *International Journal of Population Data Science* **3**:435 DOI 10.23889/ijpds.v3i3.435.

**Elias P. 2018.** The UK administrative data research network: its genesis, progress, and future. *The Annals of the American Academy of Political and Social Science* **675(1)**:184–201 DOI 10.1177/0002716217741279.

**Emms DM, Kelly S. 2019.** OrthoFinder: phylogenetic orthology inference for comparative genomics. *Genome Biology* **20(1)**:238 DOI 10.1186/s13059-019-1832-y.

**Enamorado TED, Fifield B, Imai K. 2019.** Using a probabilistic model to assist merging of large-scale administrative records. *The American Political Science Review* **113(2)**:353–371 DOI 10.1017/S0003055418000783.

**Gregg F, Eder D. 2019.** Dedupe: a Python library for accurate and scalable fuzzy matching, record deduplication and entity-resolution. GitHub. *Available at* https://github.com/dedupeio/dedupe (accessed 16 June 2020).

**Harron K, Dibben C, Boyd J, Hjern A, Azimaee M, Barreto ML, Goldstein H. 2017a.** Challenges in administrative data linkage for research. *Big Data & Society* **4(2)**:2053951717745678 DOI 10.1177/2053951717745678.

**Harron K, Hagger-Johnson G, Gilbert R, Goldstein H. 2017b.** Utilising identifier error variation in linkage of large administrative data sources. *BMC Medical Research Methodology* **17(1)**:23 DOI 10.1186/s12874-017-0306-8.

**Jordão CC, Rosa JLG. 2012.** Metaphone-pt_BR: the phonetic importance on search and correction of textual information. In: Gelbukh A, ed. *Computational Linguistics and Intelligent Text Processing, Lecture notes in computer science.* Berlin: Springer, 297–305.

**Levenshtein VI. 1966.** Binary codes capable of correcting deletions, insertions, and reversals. *Soviet Physics Doklady* **10(1965)**:707–710.

**Marcelino D. 2015.** Soundex BR: phonetic-coding for portuguese. *Available at* https://rdrr.io/cran/SoundexBR/man/soundexBR.html (accessed 23 June 2020).

**Pedregosa F, Varoquaux G, Gramfort A, Michel V, Thirion B. 2012.** Scikit-learn: machine learning in python. *The Journal of Machine Learning Research* **12**:2825–2830 DOI 10.5555/1953048.2078195.

**Perkel JM. 2021.** Ten computer codes that transformed science. *Nature* **589(7842)**:344–348 DOI 10.1038/d41586-021-00075-2.

**Pita R, Pinto C, Sena S, Fiaccone R, Amorim L, Reis S, Barreto ML, Denaxas S, Barreto ME. 2018.** On the accuracy and scalability of probabilistic data linkage over the brazilian 114 million cohort. *IEEE Journal of Biomedical and Health Informatics* **22(2)**:346–353 DOI 10.1109/JBHI.2018.2796941.

**Rognes T, Flouri T, Nichols B, Quince C, Mahé F. 2016.** VSEARCH: a versatile open source tool for metagenomics. *PeerJ* **4(17)**:e2584 DOI 10.7717/peerj.2584.

**Sariyar M, Borg A. 2010.** The recordlinkage package: detecting errors in data. *The R Journal* **2(2)**:61 DOI 10.32614/RJ-2010-017.

**Sayers A, Ben-Shlomo Y, Blom AW, Steele F. 2016.** Probabilistic record linkage. *International Journal of Epidemiology* **45(3)**:954–964 DOI 10.1093/ije/dyv322.

**Teixeira CSS, Pescarini JM, Alves FJO, Nery JS, Sanchez MN, Teles C, Ichihara MYT, Ramond A, Smeeth L, Fernandes Penna ML, Rodrigues LC, Brickley EB, Penna GO, Barreto ML, de Silva RCR. 2020.** Incidence of and factors associated with leprosy among household contacts of patients with leprosy in brazil. *JAMA Dermatology* **156(6)**:640–648 DOI 10.1001/jamadermatol.2020.0653.

**Teixeira R, Rodrigues MGA, Ferreira MD, Borges MC, Safe I, Melo GC, Spener R, Garrido MS, Monteiro WM, Siqueira AM, Lacerda MVG, Cordeiro-Santos M, de Souza Sampaio V. 2019.** Tuberculosis and malaria walk side by side in the Brazilian Amazon: an ecological approach. *Tropical Medicine & International Health* **24(8)**:1003–1010 DOI 10.1111/tmi.13282.

**Trudeau R. 2017.** Social data linkage environment. *International Journal for Population Data Science* **1(1)**:057 DOI 10.23889/ijpds.v1i1.76.

**Winkler WE. 1990.** String comparator metrics and enhanced decision rules in the Fellegi-Sunter model of record linkage. In: *Proceedings of the Section on Survey Research*. 354–369.

**Workneh MH, Bjune GA, Yimer SA. 2016.** Diabetes mellitus is associated with increased mortality during tuberculosis treatment: a prospective cohort study among tuberculosis patients in South-Eastern Amahra Region. *Ethiopia Infectious Diseases of Poverty* **5(1)**:22 DOI 10.1186/s40249-016-0115-z.