# Peer review of "Tucuxi-BLAST: Enabling fast and accurate record linkage of large-scale health-related administrative databases through a DNA-encoded approach"

_PeerJ, doi:10.7717/peerj.13507_

## Round 0.1 · original submission · Major Revisions

Please consider the Reviewers' suggestions.

·

Basic reporting

The manuscript is well written and formatted. The figures are clear and explicit. Data and software have been made available when ethically allowed.
The only typo I could see was "macOS" instead of "MacOS" in line 115

Experimental design

The methodology is sound and addresses the research question well. The methods are generally well described. The main improvement I would suggest would be a better explanation of the "codon wheel" used for encoding, and how it relates to the alphanumeric encryption key. Based on figure 1, it looks like the encryption key only needs to be a number.

Validity of the findings

The manuscript describes an innovative method for comparing and linking collections of medical records by encoding specific fields as DNA sequences and comparing them using the well-known BLAST tool. BLAST is optimised for speed and memory efficiency and allows for imperfect comparisons, which makes it a good choice for this task. The conclusions are sound and supported by the data.

Additional comments

Data protection is presented as an advantage of the new method however it seems like the encryption method used would be reasonably easy to break. However none of the other methods that Tucuxi-BLAST replaces seem to have provide encryption so data protection does not seem to be a major issue in this use case. I would suggest that the authors clarify these points and explain whether data protection is necessary in this case or whether it is just an added benefit of the method.

The method also has been optimised for names and words present in the test datasets in Brazil/Portuguese. It would be useful to know how easily the method can be adapted to other languages/countries and to provide a protocol for doing so.

Reviewer 2 ·

Basic reporting

The authors made a tool for probabilistic RL that utilizes a DNA-encoded approach to analyze and link massive administrative databases. This tool is valuable for epidemiological research in the medical field. The authors have presented a clearly written manuscript. The literature cited was sufficient. The structure, figures and tables were illustrative.
.

Experimental design

The manuscript is original since they utilize the Blastn algoritm to analyze and link massive administrative databases. The search is improved by allowing the generation of matches even with typing errors in the data bases.
The research question was well defined and relevant -yes
The rigorous investigation performed to a high technical and ethical standard- yes
Methods describes with sufficient detail and information to replicate- yes
The manuscript can be accepted for publication with revisions:
1.-Add a link for the git-hub site of Tucuxi-BLAST program availability in the introduction.
2.-When I try to run the program appear one error
./ tucuxi_blast: line1: version command not found
./tucuxi_blast: line 2: oid: command not found
This error have to be reviewed by the authors
They can be achieve this kind of problems through a forum or directly with a contact e-mail
3.- In the gitHub page you could have a more explicit help in order to know what are the input requirements and the output in a visual way

Validity of the findings

Meaningful replication encourages where rationale and benefit to literature is clearly stated- yes
The results were measure and compared with other tools availables on the web. the improvement in this tool is valuable since this tool can be used by a PC with 32 GB of RAM and not need dependenciies, which is convinient for people without eperience in linux.
Conclusions are well stated, linked to original research

Additional comments

no comment

---

## Round 0.2 · Minor Revisions

Only a few more revisions are needed

·

Basic reporting

The authors have made changes that address my previous concerns. There are a few very minor typos in the changed paragraphs though that need to be changed for readability:

Line 319: codon wheel can be easily adapted to English language -> codon wheel can be easily adapted to the English language
Line 320: correspondent -> corresponding
Line 532: Codon wheel is used -> Codon wheel used

Experimental design

All good

Validity of the findings

All good

Additional comments

The changes address my concerns. The minor typos can be fixed by the editor and only impact readability

Reviewer 2 ·

Basic reporting

No comments

Experimental design

Methods described with sufficient detail & information: in this case it should be noted that the results folder should be easily identifiable, the Readme.md file should say where they are as well as in the body of the article.

Validity of the findings

I think it is an interesting program that can be used in various areas. I think that the algorithm used is very good and its application in other areas is very interesting.

---

## Round 0.3 · accepted · Accept

The remaining minor revisions have been properly performed.